# Small Nucleolar RNA from *S. cerevisiae* Binds to Phosphatidylinositol 4,5-Bisphosphate

**DOI:** 10.3390/ncrna11040055

**Published:** 2025-07-28

**Authors:** Irma A. Jiménez-Ramírez, Miguel A. Uc-Chuc, Luis Carlos Rodríguez Zapata, Enrique Castaño

**Affiliations:** 1Unidad de Biología Integrativa, Centro de Investigación Científica de Yucatán, Calle 43, No. 130, Chuburná de Hidalgo, Mérida CP 97205, Yucatán, Mexico; anggye.apple@gmail.com; 2Laboratorio de Biología Celular, Centro de Investigaciones Regionales “Dr. Hideyo Noguchi”, Universidad Autónoma de Yucatán, Av. Itzáes, No. 490 x calle 59, Col. Centro, Mérida CP 97000, Yucatán, Mexico; ma.uc@outlook.com; 3Unidad de Biotecnología, Centro de Investigación Científica de Yucatán, Calle 43, No. 130, Chuburná de Hidalgo, Mérida CP 97205, Yucatán, Mexico; lcrz@cicy.mx

**Keywords:** snoRNA, PI(4,5)P2, *Saccharomyces cerevisiae*, lipid-binding RNA

## Abstract

**Background**: snoRNAs have traditionally been known for their role as guides in post-transcriptional rRNA modifications. Previously, our research group identified several RNAs that may bind to PIP2 with LIPRNA-seq. Among them, snR191 stood out due to its potential specific interaction with this lipid, distinguishing itself from other snoRNAs. However, a detailed study is needed to define the molecular interactions between RNA and lipids, which remain unknown but may serve as a mechanism for transport or liquid–liquid phase separation. This study aimed to determine the interaction between a snoRNA called snR191 and PIP2. **Method**: A novel methodology for RNA-PIP2 interaction was carried out. Total RNA from *Saccharomyces cerevisiae* was incubated with PIP2-bound nitrocellulose membranes and RT-PCR reactions. We performed the prediction of snR191-PIP2 interaction by molecular docking and in silico mutations of snoR191. **Results**: From LIPRNA-seq analysis, we identified that PIP2-bound RNAs were significantly enriched in diverse biological processes, including transmembrane transport and redox functions. Our RNA-PIP2 interaction approach was successful. We demonstrated that snR191 specifically interacts with PIP2 in vitro. The elimination of DNA ensured that the interaction assay was RNA-specific, strengthening the robustness of the experiment. PIP2 was docked to snR191 in a stem–loop–stem motif. Six hydrogen bonds across four nucleotides mediated the PIP2-snR191 interaction. Finally, mutations in snR191 affected the structural folding. **Conclusions**: In this study, we demonstrate the effectiveness of a new methodology for determining RNA–lipid interactions, providing strong evidence for the specific interaction between snR191 and PIP2. Integrating biochemical and computational approaches has allowed us to understand the binding of these biomolecules. Therefore, this work significantly broadens our understanding of snR191-PIP2 interactions and opens new perspectives for further research.

## 1. Introduction

Cell biology is a dynamic field constantly revealing new interactions between biomolecules that regulate vital cellular functions. RNA, a versatile biomolecule, plays diverse roles, from catalyzing biochemical reactions and storing genetic information to regulating phase separation [1,2]. Recent insights highlight RNA’s crucial role in the origin of life, supporting the RNA world hypothesis that posits RNA as a key catalyst in primitive life’s biochemical reactions [3]. In eukaryotic cells, gene expression is governed by a complex network involving both proteins and non-coding elements. Non-coding RNAs (ncRNAs), transcripts that do not encode proteins, are fundamental regulators of gene expression. Small nucleolar RNAs (snoRNAs), a class of ncRNAs historically known for guiding ribosomal RNA (rRNA) post-transcriptional modifications, exemplify this. Typically transcribed from intronic regions of protein-coding and non-protein-coding genes, these 60- to 300-nucleotide-long molecules reside within the nucleolus [4,5]. SnoRNAs are divided into C/D box snoRNAs that guide rRNA methylation and H/ACA box snoRNAs that facilitate the conversion of uridine to pseudouridine in rRNA [6]. In mammalian cells, snoRNAs have been implicated in various diseases, including cancer, neurodegenerative diseases, muscular dystrophy, and senescence [6,7,8,9].

Recent studies have shown that specific RNAs can interact with membrane lipids. For example, LINK-A is a long non-coding RNA (lncRNA) that regulates AKT activation by interacting with phosphatidylinositol 3,4,5-trisphosphate (PI(3,4,5)P3), promoting tumorigenesis and resistance to AKT inhibitors in triple-negative breast cancer [10]. Furthermore, the SELEX method has identified RNAs that interact with phosphatidylinositol 3-phosphate (PI3P), specifically an RNA aptamer [11]. It was recently discovered that small nucleolar RNA host gene 9 (SNHG9), a lncRNA widely implicated in cancer [12], binds to phosphatidic acid (PA) and LAST1 and induces liquid–liquid phase separation (LLPS) [13]. We have previously described the LIPRNAseq method, a sequencing strategy designed to detect RNAs with affinity for lipids, particularly for PIP2 [14]. Recently, Miladinovic identified an AU-rich RNA motif that can bind PIP2, a motif partially found in lncRNA HANR, which colocalizes with PIP2 in the perinuclear compartment (PNC) [15]. Furthermore, RNA has been shown to bind to ordered phospholipid bilayers, where the sequence and structure of the RNA determine the binding capacity [16]. Similarly, the interaction between RNA and phospholipid membranes can modify membrane permeability [17].

Within the large lipid family, phosphatidylinositol 4,5-bisphosphate (PI(4,5)P2 or PIP2) stands out as a central molecule in the organization of membrane domains and the regulation of both cytoplasmic and nuclear processes through its ability to interact with various proteins [18,19].

This amphipathic molecule is formed by a hydrophobic tail of fatty acids and a hydrophilic head formed by an inositol ring phosphorylated at carbon 4 and carbon 5; its synthesis is mediated by kinases and phosphatases [19]. Although PIP2 comprises 1% of membrane phospholipids [20], its function is essential in the cell as a precursor to the translation of signals and a modulator of cellular processes, such as vesicular trafficking, exocytosis, cell proliferation, cell death, and cytoskeletal dynamics [19,21]. Recently, a new type of structure called a Nuclear Lipid Island (NLI) was identified; this nuclear compartment houses PIP2 inside and RNA at its periphery, which is crucial in RNA polymerase II-dependent transcription [22]. PIP2 is also linked to the RNA polymerase I preinitiation complex and fibrillarin in the DFC region when transcription is active [23]. Therefore, PIP2 plays an important role in rRNA biogenesis.

In this context, we focused on snR191, which guides the pseudouridylation of rRNA in the nucleolus [24]. This RNA was previously identified in our LIPRNA-seq analysis as a candidate with potential affinity for PIP2, suggesting an additional lipid-mediated regulatory function. In the present study, we provide experimental evidence demonstrating that snR191 can directly bind to PIP2. To characterize this interaction, we implemented a novel methodology of incubating RNA on a nitrocellulose membrane (previously incubated with PIP2) followed by an RT-PCR reaction directly on the membrane. This methodology was developed through a collaborative effort, bringing together the expertise of researchers in RNA biology and computational modeling. Through this collective effort, we demonstrate for the first time the interaction between snR191 and PIP2, a discovery that sheds new light on the role of these two biomolecules in cellular processes. These two biomolecules interact through hydrogen bonds between the adenine and uracil residues of snR191 and the oxygen atoms of PIP2.

## 2. Results

### 2.1. PIP2-Associated RNAs Are Enriched in Cellular Transport and Metabolism Processes

We identified these RNAs using the Lipid-interacting RNA sequencing (LIPRNAseq) method developed by our research group [14]. In this approach, we used total RNA from *S. cerevisiae* and performed an RNA pulldown with PIP2-conjugated sepharose beads, followed by sequencing. This strategy allowed us to detect PIP2-associated RNAs enriched in various biological processes and molecular functions (Figure 1). 

As shown in Figure 1a, the analysis of biological processes revealed the most enriched terms for transmembrane transport, vesicle-mediated transport, and organophosphate metabolic processes, suggesting an essential role for these RNAs in regulating molecular trafficking and cellular homeostasis. Additionally, we identified terms linked to mitochondrial organization and energy metabolism, indicating a potential role of PIP2-associated RNAs in bioenergetic regulation.

Figure 1b displays the molecular functions, highlighting significant enrichment in transmembrane transporter activity, transporter activity, and structural molecule activity, which suggests a role in ion and molecule transport across membranes. We also observed an enrichment in oxidoreductase activity, pointing to the possible involvement of these RNAs in redox processes. These results suggest that RNAs that interact with PIP2 are associated with genes involved in key functions related to transport processes, metabolism, and structural stability, providing a possible functional relevance of these RNAs in the cell.

### 2.2. snRNA 191 Specifically Binds PIP2 In Vitro

In previous studies, our research group developed LIPRNAseq, a method for sequencing RNAs that interact with lipids, specifically PIP2 [14]. Using our LIPRNA-seq-generated database, we identified various types of RNAs, including snR191 from *S. cerevisiae*. To confirm that snR191 can interact with PIP2 in vitro, we first cultured *S. cerevisiae* cells and isolated total RNA using acidic phenol. We performed DNase Zero treatments to ensure the removal of DNA residues (Figure 2a) that could be a potential contaminant for the RNA–lipid interaction assays. PCR analysis of treated and untreated RNA showed no amplification in either case, indicating a complete removal of gDNA (Figure 2b).

Subsequently, we performed RT-PCR using the same RNA samples (with and without DNase Zero treatment). The results showed the amplification of a 274 bp fragment corresponding to snR191 under both conditions, suggesting that reverse transcription was efficient in both cases (Figure 2c).

We then carried out a lipid binding assay on a nitrocellulose membrane impregnated with 100 pMol of PIP2 (Figure 2d).

As illustrated in Figure 2d (step 3), we supplemented the membrane with bacterial RNA as a source of non-specific RNA. This step was designed to test whether snR191 binding to PIP2 could be competitively displaced by unrelated RNA molecules, thus evaluating the specificity of the interaction.

After blocking the membrane and incubating it with total RNA (step 5), the unbound fraction (flow-through, FT) was collected during step 6 by recovering the supernatant after the centrifugation of the incubation mixture. The bound fraction (B) was assessed after the washing steps (step 7) by performing RT-PCR directly on the membrane to detect membrane-retained RNAs, particularly snR191. A membrane without PIP2 was used as a negative control (Figure 2e). Only in the presence of PIP2 did we detect RNA predominantly in the bound fraction (B), indicating a specific interaction.

### 2.3. Three-Dimensional Modeling of S. cerevisiae snR191, Multiple Sequence Alignment, and Motif Analysis

The biological functions of non-coding RNAs are highly dependent on their native 3D structure and complex formation with ligands [25]. In this study, we modeled the 3D structure of *S. cerevisiae* snR191 (Figure 3a,b) to determine structural folding characteristics and identify possible PIP2-binding motifs. snR191 is packed in a relaxed helical manner, is 274 nucleotides long, and consists primarily of uracil residues (100 nt). Secondary structures, such as double-stranded helices, hairpins, and single-stranded loops, are connected by tertiary interactions. Double-stranded helices are stabilized by Watson–Crick base pair bonding (Figure 3a and Appendix A), and tertiary contacts are typically formed by divalent ions such as magnesium [26]. The structure of snR191 is composed of a multi-branch loop, a unique tetraloop motif (UUGG), and a H/ACA motif associated with site-specific pseudouridylation for RNA [27] (Appendix A). Multiple alignments revealed that all four snoRNAs analyzed in this study, including snR191, are conserved with *LINK-A* (Figure 3d). A fifty-nt region of snR191 was the best-aligned sequence with *LINK-A* (Figure 3c,d). Furthermore, we identified twenty-two protein-binding motifs in snR191 (Appendix A) in silico, but no PIP2-binding motifs. These results suggest that snR191 is conserved with *LINK-A* (in the fifty-three nt sequence alone) and may play a multifunctional role.

### 2.4. Molecular Docking

Our in vitro results showed the snR191-PIP2 interaction. The next step was to determine the interaction mode in silico. To find out, we performed molecular docking experiments (Figure 4a). The docking results showed that PIP2 physically binds to the stem–loop motif of snR191 (Figure 4b). The binding stability of snR191-PIP2 was mediated by the formation of six hydrogen bonds across four nt of snR191 (Figure 4c,f and Appendix A) located in the region comprising the fifty nt sequence (Figure 4d,e). The snR191-PIP2 interaction occurs through two hydrogen bonds between residues A31 with the oxygen-11 atom of the 2′-phosphate and A45 with the oxygen-4 atom of the 1′-phosphate of PIP2, whereas A43 and U44 form four H-bonds with the oxygen-16 and oxygen-17 atoms of PIP2 inositol, respectively (Figure 4c). However, the snR191-PIP2 complex may bind to other nearby nt through tertiary interactions of the snR191 structure, as shown in Figure 4. Although the predicted theoretical affinity of snR191-PIP2 was very high (Figure 4e) due to the docking program used, it needs to be experimentally validated. The structure and chemical composition determine the function of lncRNAs, and a crucial feature is their propensity to fold into thermodynamically stable secondary and higher-order structures [28]. Therefore, we performed in silico experiments to investigate whether site-specific mutations affect the folding of snR191 (Appendix A). Single-nucleotide mutations, A43G, and triple-nucleotide mutations, A43U-U44A-A45U, dramatically affected the folding snR191 compared to the wild type (Appendix A). Furthermore, deleting the 50-nt region in snR191 also led to the observation of a folding change (Appendix A). Our in silico results suggest that mutations at positions A43-U44-A45 of snR191 alter its folding relative to the WT and impair its binding to PIP2.

## 3. Discussion

In this study, we used the LIPRNAseq technique [16], developed by our group, to identify a set of PIP2-associated RNAs. Functional enrichment analysis revealed a significant overrepresentation of genes involved in key processes, including transmembrane transport, vesicular trafficking, organophosphate metabolism, and energy catabolism. These findings are consistent with PIP2’s well-established role as a lipid regulator in membrane dynamics, including exocytosis, endocytosis, and cytoskeletal assembly [29,30]. The association of certain RNAs with PIP2 suggests a possible additional regulatory function, in which these RNAs could contribute to the spatial and temporal organization of these processes. The enrichment of molecular functions, such as transporter and oxidoreductase activity, indicates that PIP2-associated RNAs may be involved in regulating ion and metabolite exchange and maintaining redox balance—key processes for cellular homeostasis and energy metabolism [31,32]. Specifically, the detection of snR191 within the PIP2-associated RNA pool raises questions about non-canonical functions of this snoRNA, given its role in rRNA modification in the nucleolus—a region rich in RNA–protein interactions and susceptible to phase-separation organization [33]. The implication of transport and organization functions suggests that certain RNAs, such as snR191, could participate in nucleolar dynamics facilitated by lipids like PIP2. On the other hand, the experimental validation that snR191 specifically binds to PIP2 in vitro provides direct evidence of physical interaction between an ncRNA and a lipid, an emerging field of study that has attracted significant interest in recent years [10,13,14]. Our methodology guarantees the removal of genomic DNA and ensures the specificity of the observed signal, allowing a robust assessment of the interaction. PIP2-impregnated membrane binding assays and RT-PCR detection showed that snR191 is mainly found in the bound fraction only in the presence of PIP2. These results are consistent with those reported by Lin et al., who used overlay assays using lipid membranes and biotin-labeled lncRNA Link-A, demonstrating the interaction with PIP2 and PIP3 [10]. Although we did not use multi-phosphoinositide lipid strips and did not label our RNA, our results have been favorable. The RT-PCR-based approach after RNA incubation represents a novel tool for the in vitro characterization of these types of interactions and could be applied to other candidate RNAs in future studies. This observation represents a novel finding, considering that snR191 is a yeast snoRNA that has so far been well characterized mainly for its role in post-transcriptional modifications of rRNA [34,35]. The ability of snR191 to bind PIP2 suggests that this snoRNA may have functions beyond its role in rRNA modification. An intriguing possibility is that such an interaction contributes to the spatial organization of ribonucleoprotein complexes through the liquid–liquid phase separation mechanism. Recent studies have shown that specific ncRNAs can promote the formation of biomolecular condensates through interactions with lipids. For example, the lncRNA SNHG9 binds phosphatidic acid (PA) to induce the liquid–liquid phase separation of the LAST1 protein, modulating the Hippo signaling pathway [13]. Additionally, the association of PIP2 with RNApoII has been observed to induce the formation of nuclear condensates, directly impacting transcription [36].

Three-dimensional modeling showed that the architecture of snR191 comprises secondary structures such as double-stranded helices, hairpins, and single-stranded loops (Appendix A). We identified a tetraloop motif (UUGG) with a Watson–Crick U-A closure base pair. Tetraloop-forming sequences have been reported to be conserved and play a role in rRNA’s structural stability and functionality [37]. Furthermore, snR191 contains an H/ACA box located three nucleotides upstream of the 3′ end (Appendix A). Previous studies have documented that snoRNAs with an H/ACA motif guide the pseudouridylation of the large subunit rRNA at positions U2258 and U2260 and are highly conserved in eukaryotes [24]. Multiple alignment analysis showed that all four *S. cerevisiae* snRNAs aligned with LINK-A, including snR191. The region best aligned with LINK-A corresponded to a fifty-nucleotide sequence, of which seven nucleotides are highly conserved (Figure 3c,d). Interestingly, the 1081–1140 nt region of LINK-A has been described as responsible for lipid binding. In this region, weak binding to PIP2 and minimal interaction with other PIPs was found [10]; this region comprises fifty-nine nucleotides. Consistently, the fifty- and fifty-nine nt sequences of snR191 and LINK-A, respectively, were the ones that aligned (Figure 3d). On the other hand, of the twenty-two motifs we identified in snR191 (Appendix A), none correspond to lipid binding. This is because efficient bioinformatics tools for RNA–lipid interactions have not been developed. However, we found repeated motifs (twice) for binding to T-cell restriction intracellular antigen 1 (TAI1), a cytotoxic granule-associated RNA-binding protein in humans. The low complexity domain (LCD) of TIA1 plays a critical role in stress granule assembly through liquid–liquid phase separation and is also involved in the regulation of alternative pre-RNA RNA splicing and mRNA translation by binding to U-rich RNA sequences [38,39,40].

Techniques such as X-ray crystallography and nuclear magnetic resonance (NMR) have helped elucidate RNA’s three-dimensional structure [41]. However, these tools only provide snapshots of a given moment under specific conditions, and their information on cellular dynamics, such as RNA–lipid interactions, is limited. In this study, we performed molecular docking experiments that predicted the snR191-PIP2 interaction (Figure 4) and showed that site-specific mutations can affect structural folding (Appendix A). Previous studies have shown that LncRNA LINK-A physically interacts with the lipid PIP3 with a binding affinity of kd: 141 nM and that the 3′-phosphate of PIP3 is important for LINK-A binding [10]. Our results showed that the snR191-PIP2 interaction occurs via six hydrogen bonds, two H-bonds between residues A31 and the oxygen-11 atom of PIP2 2′-phosphate, and A45 with the oxygen-4 atom of PIP2 1′-phosphate. Meanwhile, A43 and U44 form four H-bonds with the oxygen-16 and oxygen-17 atoms of PIP2 inositol, respectively (Figure 4c). However, it is possible that the snR191-PIP2 complex interacts with other nearby nts due to the tertiary structures of snR191, as shown in Figure 4c. While the results suggest a direct interaction between snR191 and PIP2, it is important to validate this binding using other interaction methods that provide a more robust characterization. Techniques such as Surface Plasmon Resonance would allow for the determination of the interaction parameters between snR191 and PIP2. Complementarily to this, the application of isothermal titration calorimetry (ITC) could provide more detailed information on the thermodynamics of this association, allowing a deeper understanding of the binding mechanism and the physicochemical nature of the snR191-PIP2 complex.

Furthermore, the implementation of targeted mutational analysis in specific regions of snR191 potentially involved in the interaction with PIP2, combined with binding assays, would allow for the direct validation of the interaction [10]. This approach would distinguish between residues that are important for structure or function, thus reinforcing the specificity of snR191 and PIP2. Together, these studies would provide more detailed information on the interaction model between snR191-PIP2, opening a new hypothesis of a lipid-mediated regulatory function of snR191.

## 4. Materials and Methods

### 4.1. Gene Set Enrichment Analysis

We used sequencing data previously published by our group [14] for functional enrichment analysis. Gene set enrichment analysis (GSEA) was performed in R software (version 4.4.2) using the ClusterProfiler package (version 4.14.4). Differentially expressed genes (DEGs) were used, both up- and down-regulated. Functional annotations were based on Gene Ontology (GO) terms for Biological Processes and Molecular Functions.

### 4.2. Cell Culture of S. Cerevisiae

*S. cerevisiae* cells (S288C strain) were cultured in 50 mL of YPD growth medium (1% yeast extract, 2% peptones, 2% glucose, pH 6.5) at 25 °C with constant shaking (200 rpm) overnight to ensure adequate oxygenation and culture homogeneity. Cells were initially inoculated at a density of 0.1 OD600 and grown until reaching an optical density of 0.6–0.8 OD600; at this point, they were harvested for subsequent experiments.

### 4.3. Total RNA Extraction from S. cerevisiae

Total RNA was isolated from *S. cerevisiae* cells using an acid phenol extraction protocol with some modifications. Cells were harvested by centrifugation at 4000× *g* for 5 min at 4 °C and then frozen at −80 °C for 2 h. Subsequently, cells were resuspended in 400 µL of TES solution (10 mM Tris-HCl, pH 7.5; 10 mM EDTA; 0.5% SDS) and 400 µL of acid phenol (pH 4.3). The mixture was homogenized by vortexing for 15 min. The sample was incubated at 60 °C in a heat block for 5 min. Afterward, it underwent a freeze–thaw cycle in liquid nitrogen for 5 min, followed by centrifugation at 15,000× *g* for 10 min at 4 °C. The supernatant was transferred to a 1.5 mL microcentrifuge tube, and 400 µL of acid phenol (Merck KGaA, Darmstadt, Germany) was added. The mixture was vortexed for 15 min and then centrifuged at 15,000× *g* for 10 min at 4 °C. The supernatant was transferred to a new 1.5 mL microtube, and 400 µL of chloroform was added. The mixture was vigorously shaken for 10 min and centrifuged at 15,000× *g* for 10 min at 4 °C. The supernatant was transferred to a new, RNase-free 1.5 mL microtube, and the RNA was precipitated by adding 40 µL of 3 M sodium acetate and 1000 µL of ice-cold absolute ethanol (molecular biology grade). The mixture was centrifuged at 15,000× *g* for 10 min at 4 °C. The RNA pellet was washed with 70% ice-cold ethanol and resuspended in nuclease-free water. The concentration and integrity of the total RNA were assessed by spectrophotometry (260/280 nm) and electrophoresis on a 1% agarose gel, respectively.

### 4.4. DNase Treatment for Removal of Contaminating DNA

The RNA obtained was treated with Baseline-ZERO™ DNase (Middleton, WI, USA) to eliminate potential DNA contamination. The treatment followed the manufacturer’s recommendations. The DNase-treated total RNA was further purified by phenol–chloroform extraction. To visualize DNA removal and RNA integrity, a 1% agarose gel was run.

### 4.5. RNA–Lipid Interaction Assay

A protocol was designed to evaluate the interaction between RNA and lipids using nitrocellulose membranes and employing the RT-PCR technique. The nitrocellulose membranes (Little Chalfont, Buckinghamshire, UK) were cut into 1 cm^2^ squares, inserted into 0.2 mL PCR microtubes, and sterilized in an autoclave at 121 °C for 18 min. PIP2 (Avanti Polar Lipids, Alasbaster, AL, USA) was dissolved in a chloroform/methanol/water mixture (20:9:1, *v*/*v*) for application onto nitrocellulose membranes. Two microliters of a solution containing 100 pmoles (pMol) of PIP2 were spotted directly onto the membrane, forming a dot approximately 4–5 mm in diameter. The membrane was then dried under vacuum for 30 min. Subsequently, the membranes were washed with 100 µL of RNA–lipid binding buffer (50 mM HEPES, pH 7.0; 50 mM NaCl) as described by Lin et al. [10], centrifuged in a DLAB minicentrifuge model D1008 at 2680× *g* for 30 s, and the supernatant was discarded. Membranes were then blocked with RNA–lipid binding buffer supplemented with 10 ng/µL bacterial RNA at 4 °C for 25 min. Centrifugation was carried out at 2680× *g* for 30 s, and the supernatant was discarded. Nitrocellulose membranes were incubated with 100 µL of RNA–lipid binding buffer supplemented with 180 ng/µL yeast RNA and 1 µL of RNase inhibitor SUPERase. In (Invitrogen, Graiciuno, Vilnius, Lithuania) at 4 °C for 25 min. They were then centrifuged at 2680× *g* for 30 s to collect the unbound fraction. Nitrocellulose membranes were washed three times with 100 µL of RNA–lipid binding buffer containing 0.1% Nonidet P-40 (NP-40) and centrifuged at 2680× *g* for 30 s. Finally, RT-PCR was performed.

### 4.6. RT-PCR of snR191 Bound to Nitrocellulose Membrane with PI(4,5)P2

To find out whether snR191 interacts with PIP2, RT-PCR experiments were performed using the nitrocellulose membranes from the RNA–lipid interaction assays described above as a template. RT-PCR was performed in 0.2 mL tubes containing the RNAs bound to the nitrocellulose membrane (Figure 2d). For cDNA synthesis, the following reagents were used: 1 µL of snR191-specific reverse oligo, membrane-bound RNA, 1 µL of 10 mM dNTP mix, nuclease-free water, 4 µL of 5X buffer, 2 µL of 0.1 M DTT, 1 µL of RNase inhibitor SUPERase. In (20 U/µL), and 1 µL of M-MLV enzyme (200 U/µL). The reaction conditions were 65 °C for 5 min, 37 °C for 2 min, and 37 °C for 50 min. PCR carried out cDNA amplification in a final volume of 25 µL, using primers specific to snoR191. The PCR reaction was performed in a thermal cycler (TECHNE TC-512 (from keison.co.uk) with the following conditions: initial denaturation at 95 °C for 1 min, followed by 28 cycles of denaturation at 95 °C for 30 s, annealing at 65 °C for 30 s, and extension at 72 °C for 1 min. The amplified products were visualized by 3% agarose gel electrophoresis stained with SYBR Safe (Invitrogen, Carlsbad, CA, USA). The specific forward primers—CGCGATTACCAAACCTTTTTGTC (Tm: 67.4 °C)—and reverse—AATTGTGAGGATCTTTACTACG (Tm: 57 °C)—of snR191 were designed manually and verified using the online software Oligocal (http://oligocalc.eu/; accessed on 25 April 2024).

### 4.7. Multiple-Sequence Alignment

The multiple sequence alignment was carried out with MUSCLE (Multiple Sequence Comparison by Log-Expectation) software at https://www.ebi.ac.uk/Tools/msa/muscle/ (EMBL-EB, Hinxton, UK; accessed on 11 May 2024), using default parameters. The nucleotide sequences of snR11, snR34, snR63, and snR191 were used for the alignment, and LINK-A (accession: LINC01139) was used as the reference. For the motif search analysis for snR191, the following freely available web server was used: http://brio.bio.uniroma2.it (accessed on 30 May 2024).

### 4.8. Molecular Docking

Molecular docking prediction was performed to evaluate the binding properties of PIP2-snR191 using the HDOCK server (http://hdock.phys.hust.edu.cn/; accessed on 21 January 2025). Three-dimensional (3D) snR191 and PIP2 structures were constructed in PDB format. The 3D structure of snR191 was built from the nucleotide sequence using the RNAcomposer online server (https://rnacomposer.cs.put.poznan.pl/; accessed on 21 January 2025). RNAComposer is a highly effective tool for predicting 3D RNA structures, even for non-coding RNAs up to 500 nucleotides. It uses a combination of thermodynamic principles and fragment assembly, ensuring that base-pairing interactions are thermodynamically correct. RNAComposer maps secondary structures against a database of RNA motifs to build the final 3D model. This server generates accurate 3D models of complex structures such as ribozymes, riboswitches, and large non-coding RNAs [42,43,44]. The 3D structure of PIP2 was built with the molecular smile using the built structure editing option of the UCSF Chimera 1.14 Molecular Graphics Systems software [45]. The PIP2 smile was downloaded from the PubChem database (https://pubchem.ncbi.nlm.nih.gov/) (accessed on 7 January 2025). The docking scoring function is based on the MMFF94S force field. Force field-based functions consist of a sum of energy terms from a classical force field, usually considering the receptor–ligand complex’s interaction energies and the ligand’s internal energy. For standard docking, a blind molecular docking grid was used to define the docking region. This was carried out because the specific PIP2-snR191 binding sites are unknown. The parameters are called default values on the server. All results were visualized using UCSF Chimera 1.14 Molecular Graphics Systems software.

## 5. Conclusions

This study demonstrates that snR191 specifically binds to PIP2 using a novel methodology that combines RNA–lipid interaction assays and molecular docking analysis. The interaction of snR191 and PIP2 suggests a mechanism for regulating post-transcriptional modifications of rRNA. However, further studies will be needed to clarify the functional consequences of this interaction. Taken together, our findings open a new line of research into the role of this RNA–lipid interaction. Unlocking the complexities of lipid–RNA–protein interactions is crucial for deciphering gene regulation mechanisms, which is particularly important since recent technological advancements have revealed the extensive processing and vast diversity of RNA species within cells.

## Figures and Tables

**Figure 1 ncrna-11-00055-f001:**
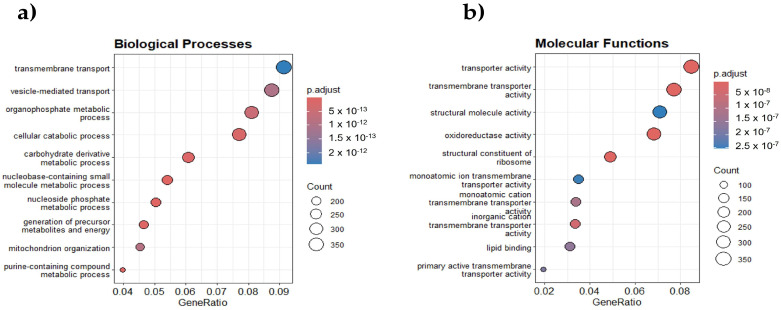
Gene enrichment analysis (GSE) using LIPRNAseq. (**a**) Significantly enriched biological processes, including transmembrane transport, organophosphate metabolism, and mitochondrial organization. (**b**) Enriched molecular functions. The size of the dots indicates the number of genes, while the color reflects statistical significance (p.adjust).

**Figure 2 ncrna-11-00055-f002:**
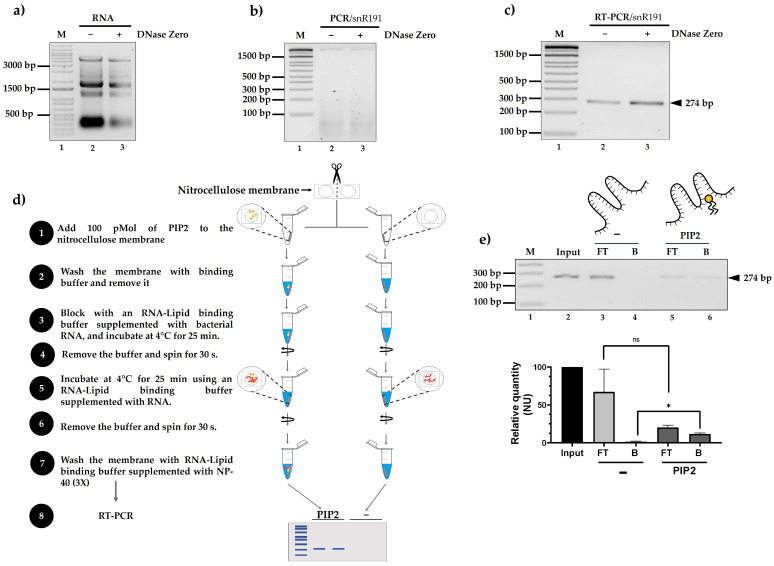
Interaction of snR191 with PIP2 in vitro. (**a**) DNase Zero-treated and untreated RNA. (**b**) PCR of snR191 in samples with and without DNase treatment. (**c**) RT-PCR specific amplification of snR191, confirming transcript presence. (**d**) Flowchart of the methodology used to assess RNA–lipid interaction. In 3, bacterial RNA was supplemented as a non-specific competitor to assess the specificity of snR191-PIP2 binding. (**e**) snR191–PIP2 interaction assay. Detection of snR191 in unbound (FT) and bound (B) fractions by RT-PCR. The FT fraction was collected in step 6, after total RNA incubation (step 5), by recovering the supernatant. The B fraction was obtained in step 7 after the washes by performing RT-PCR directly on the membrane. A membrane without PIP2 was used as a negative control. We quantified the signal using ImageJ-win64 (2019) software and represented it as relative quantity (normalized units, NU). The Y-axis indicates the relative amount of snR191 (in NU). Statistical differences were determined using Student’s *t*-test. A *p*-value < 0.05 was considered statistically significant. Statistical significance is indicated in the figure as follows: * *p* < 0.05; ns = not significant. Input: corresponds to total RNA before incubation with the membrane, FT: flow-through (RNA that did not bind), B: bound (RNA that did bind), PIP2: phosphatidylinositol 4,5-bisphosphate.

**Figure 3 ncrna-11-00055-f003:**
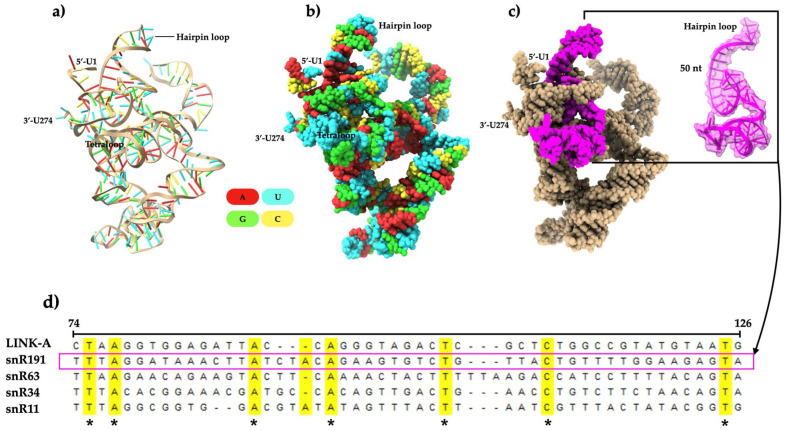
Three-dimensional structural modeling of the *S. cerevisiae* snR191 and multiple alignments of snR191, snR163, snR43, snR11, and LINK-A. (**a**) Ribbon-like 3D structure. (**b**) Surface view. (**c**) The surface view in which the 3D structure of the sequence corresponding to fifty is not shown in panel (**d**) is highlighted in magenta and marked within a rectangle. (**d**) Multiple alignments of nucleotide sequences of four snoRNAs, with LINK-A as a reference. Asterisks (*) indicate conserved nt.

**Figure 4 ncrna-11-00055-f004:**
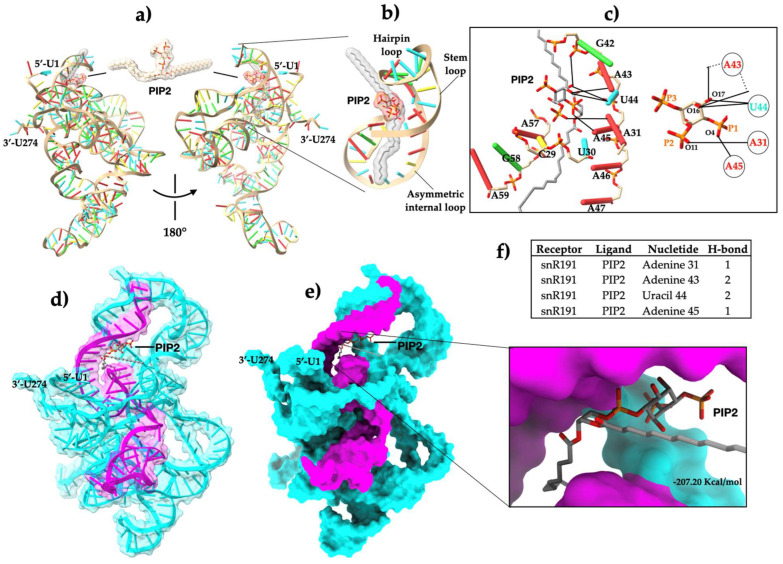
Molecular docking of the snR191-PIP2 complex. Panel (**a**), snR191-PIP2 binding. Panel (**b**), PIP2 pose in the stem–loop motif. Panel (**c**) shows a view of the nt interacting with PIP2 and potential hydrogen bonds in solid lines. Panel (**d**), transparent surface view of PIP2 docked with snR191. Panel (**e**), surface view of snR191-PIP2 docking and close-up of the PIP2 pose; the binding affinity energy was −207.20 Kcal/mol. Panel (**f**), nucleotides interacting with PIP2 through six H-bonds. In snR191, red indicates adenine, cyan uracil, green guanide, and yellow cytokinin. In PIP2, red indicates oxygens (O), orange phosphates, and gray carbon atoms. The 50-nt sequence is shown in magenta. All images were generated using the UCSF Chimera 1.14 software.

## Data Availability

The data used were previously published [14] under the accession numbers for BioSamples are SAMN33417987-Sc_total, SAMN33417988-Sc_control, SAMN33417989-Sc_PIP2.

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
