# Peer review of "Small Nucleolar RNA from S. cerevisiae Binds to Phosphatidylinositol 4,5-Bisphosphate"

_ncrna, 2025, doi:10.3390/ncrna11040055_

Round 1
Reviewer 1 Report
Comments and Suggestions for Authors
The manuscript titled "Small nucleolar RNA from S. cerevisiae binds to Phosphatidylinositol 4,5-bisphosphate" presents a novel investigation into the interaction between snR191 and PIP2, combining biochemical assays and computational modeling. The study demonstrates a specific RNA-lipid interaction, supported by in vitro experiments and molecular docking, suggesting potential roles for snoRNAs beyond rRNA modification. The methodology is innovative, and the findings could open new avenues in RNA-lipid research.
However, several issues need addressing to strengthen the manuscript.
1.The 3D model of snR191 is theoretical. Validate predictions with experimental data (e.g., SHAPE-MaP or enzymatic probing) or cite precedent studies using RNAcomposer for similar RNAs.
2.The concentration of PIP2 used can be further described.
3.“snRNA191” and “snR191” can be used interchangeably (e.g., summary vs. results),and should perhaps be named consistently.
Author Response
Reviewer 1)
The manuscript titled "Small nucleolar RNA from S. cerevisiae binds to Phosphatidylinositol 4,5-bisphosphate" presents a novel investigation into the interaction between snR191 and PIP2, combining biochemical assays and computational modeling. The study demonstrates a specific RNA-lipid interaction, supported by in vitro experiments and molecular docking, suggesting potential roles for snoRNAs beyond rRNA modification. The methodology is innovative, and the findings could open new avenues in RNA-lipid research.
However, several issues need addressing to strengthen the manuscript.
- The 3D model of snR191 is theoretical. Validate predictions with experimental data (e.g., SHAPE-MaP or enzymatic probing) or cite precedent studies using RNAcomposer for similar RNAs.
ANSWER: Thank you very much for the comment. We agree with this comment. Therefore, in the materials and methods section, we have added the following paragraph and citations (line 424-431).
“RNAComposer is a highly effective tool for predicting 3D RNA structures, even for non-coding RNAs up to 500 nucleotides. It uses a combination of thermodynamic principles and fragment assembly, ensuring that base-pairing interactions are thermodynamically correct. RNAComposer maps secondary structures against a database of RNA motifs to build the final 3D model. This server generates accurate 3D models of complex structures such as ribozymes, riboswitches, and large non-coding RNAs (Biesiada et al., 2016; Leitão & Enguita, 2025; Purzycka et al., 2015).”
Biesiada, M., Pachulska-Wieczorek, K., Adamiak, R. W., & Purzycka, K. J. (2016). RNAComposer and RNA 3D structure prediction for nanotechnology. Methods, 103, 120-127. https://doi.org/https://doi.org/10.1016/j.ymeth.2016.03.010
Leitão, A. L., & Enguita, F. J. (2025). The Unpaved Road of Non-Coding RNA Structure-Function Relationships: Current Knowledge, Available Methodologies, and Future Trends. Noncoding RNA, 11(2). https://doi.org/10.3390/ncrna11020020
Purzycka, K. J., Popenda, M., Szachniuk, M., Antczak, M., Lukasiak, P., Blazewicz, J., & Adamiak, R. W. (2015). Automated 3D RNA structure prediction using the RNAComposer method for riboswitches. Methods Enzymol, 553, 3-34. https://doi.org/10.1016/bs.mie.2014.10.050
- The concentration of PIP2 used can be further described.
ANSWER: We thank the reviewer for this comment. We have updated the Materials and Methods section to include a more detailed description of the application procedure and the amount of PIP2 used (Lines 373 - 377).
- “snRNA191” and “snR191” can be used interchangeably (e.g., summary vs. results), and should perhaps be named consistently.
ANSWER: Thank you for pointing out this inconsistency. We have revised the manuscript and now consistently use the “snR191” nomenclature throughout.
Reviewer 2 Report
Comments and Suggestions for Authors
Manuscript ncrna-3687694 by Irma A et al. investigates the roles of snRNA in various biological processes through application of LIPRNA-seq analysis and performed conceptual characterization for the binding of a specific snRNA (snR191) to PIP2 with both experimental method and computational docking analysis to unfold the binding interactions at molecular level. These characteristics demonstrated the feasibility of the binding of snRNA to PIP2 in vitro and in silico, suggesting the implications for further research.
This manuscript provided initial non-quantitative characterization of the binding between snR191 and PIP2, I have minor comments below:
Introduction lines 143-145 and Figure 2: it is recommended that authors specify at which step FT and B fractions were collected, e.g. after step 5 to collect FT and maybe add annotates in Figure 2 to clarify the collection of both fractions.
Discussion: maybe it's interesting to discuss the characterization of snRNA binding to PIP2 using orthogonal methods in vitro. Additionally, investigating the binding affinity of snRNA with PIP2 through techniques such as Biacore could provide insights into the strength of this interaction.
Author Response
Manuscript ncrna-3687694 by Irma A et al. investigates the roles of snRNA in various biological processes through application of LIPRNA-seq analysis and performed conceptual characterization for the binding of a specific snRNA (snR191) to PIP2 with both experimental method and computational docking analysis to unfold the binding interactions at molecular level. These characteristics demonstrated the feasibility of the binding of snRNA to PIP2 in vitro and in silico, suggesting the implications for further research.
This manuscript provided initial non-quantitative characterization of the binding between snR191 and PIP2, I have minor comments below:
- Introduction lines 143-145 and Figure 2: it is recommended that authors specify at which step FT and B fractions were collected, e.g. after step 5 to collect FT and maybe add annotates in Figure 2 to clarify the collection of both fractions.
ANSWER: We thank the reviewer for the suggestion. As requested, we have now clarified in the text that the unbound fraction (FT) was collected during step 6, following RNA incubation (step 5), and that the bound fraction (B) was assessed after the washing step (step 7) by performing RT-PCR directly on the membrane. These details are now explicitly described in the revised manuscript (Lines 148 - 153), and we have also annotated Figure 2 accordingly to indicate the collection points of both FT and B fractions.
- Discussion: maybe it's interesting to discuss the characterization of snRNA binding to PIP2 using orthogonal methods in vitro. Additionally, investigating the binding affinity of snRNA with PIP2 through techniques such as Biacore could provide insights into the strength of this interaction.
ANSWER: We appreciate this suggestion. We agree that the use of other methods would be useful to validate and further characterize the interaction between snR191 and PIP2. In particular, techniques such as surface plasmon resonance could provide quantitative information on binding affinity, which would complement our findings. Furthermore, we have incorporated several techniques that would allow us to gain more detailed information about this interaction. Therefore, we have included a paragraph in the Discussion section of this paper to reflect on this possibility and highlight it as a relevant direction for future studies. (Lines 311-325)
Reviewer 3 Report
Comments and Suggestions for Authors
The manuscript presents a compelling novel finding regarding the interaction between snR191 and PIP2, supported by both experimental and computational analyses. To further enhance its quality and impact, the following revisions are recommended:
- Standardize the abbreviation for snR191 throughout the manuscript, ensuring a single, consistent usage (e.g., snR191).
- The abstract could briefly mention the specific snRNA (snR191) identified by LIPRNA-seq, as it's the focus of the paper.
- While the introduction provides a good background on snoRNAs and PIP2, the transition from general RNA-lipid interactions to the specific focus on snR191 could be smoother.
- The description of Fig. 1 is good but emphasizing why these enriched processes/functions are relevant to snR191 or RNA-lipid interactions in general would strengthen the discussion of these results.
- While the methodology is novel, Figure 2d could benefit from a clearer explanation of what is the function of the supplemented bacterial RNA step 3.
- In Figure 2e, what does “Input" represents in the gel image? Is it the total RNA before the assay? The bar graph needs clear units on the y-axis (NU is defined, but it might be clearer with "Relative Quantity" or similar, along with NU). it states "p−value<0.05, ns= not significant" for the statistical analysis here, please ensure that the statistical test (t-test) and significance levels are clearly indicated for all comparisons shown in the graph. Please correct the inconsistent capitalization of "Pi(4,5)P2" in Figure.
- Please clarify the selection process for snR191 among the four cerevisiae snRNAs shown in Figure 3d as being associated with PIP2, providing context for its specific inclusion.
- Please correct the inconsistent values in Figure 4e and the figure legends. The binding affinity value is very high. While docking scores can be high, it would be beneficial to provide context or a brief interpretation of this value in relation to known RNA-ligand interactions, or mention that these are theoretical predictions that require experimental validation of affinity.
- Please rephrase the frank statement in lines 287-290. While acknowledging the study's limitations, it would be more impactful to propose specific experimental approaches (e.g., mutational analysis combined with binding assays, isothermal titration calorimetry) that would validate the predicted interaction mode, demonstrating a comprehensive understanding of the necessary next steps.
- Typographical errors and grammatical issues: description of Table S1 in line 174-175, title of Figure 3d, and hypos “Figure 64” in line 197, "Fig. 1d" in line 248 and "Fig. 1e" in line 249, “Figure 6c” in line 287.
Author Response
The manuscript presents a compelling novel finding regarding the interaction between snR191 and PIP2, supported by both experimental and computational analyses. To further enhance its quality and impact, the following revisions are recommended:
- Standardize the abbreviation for snR191 throughout the manuscript, ensuring a single, consistent usage (e.g., snR191).
ANSWER: Thank you for pointing out this inconsistency. We have revised the manuscript and now consistently use the "snR191" nomenclature throughout.
- The abstract could briefly mention the specific snRNA (snR191) identified by LIPRNA-seq, as it's the focus of the paper.
ANSWER: We appreciate the suggestion. On lines 18-19, we have modified the abstract to include explicit mention of snR191 from the background, highlighting its identification by LIPRNA-seq and its relevance as a primary focus of the study.
- While the introduction provides a good background on snoRNAs and PIP2, the transition from general RNA-lipid interactions to the specific focus on snR191 could be smoother.
ANSWER: We've made the changes, improving the flow. In this new paragraph, we highlight that snR191, a snoRNA that guides pseudouridylation of rRNA in the nucleolus, was identified in our LIPRNA-seq analysis as a candidate with affinity for PIP2. This observation suggests that snR191 may have an additional lipid-regulated function, which prompted its selection as a model for studying RNA-PIP2 interactions (lines 89 – 92).
- The description of Fig. 1 is good but emphasizing why these enriched processes/functions are relevant to snR191 or RNA-lipid interactions in general would strengthen the discussion of these results.
ANSWER: We appreciate the comment. We have expanded the discussion associated with Figure 1 to contextualize the biological relevance of the observed enriched processes. In particular, we highlight how functions such as transmembrane transport, redox metabolism, and RNA organization could be related to the role of snR191 in the nucleolus and its potential functional interaction with PIP2, suggesting a possible involvement in lipid-mediated subcellular organization or membraneless compartmentalization mechanisms. This interpretation has been incorporated in lines 246–250 of the Discussion.
- While the methodology is novel, Figure 2d could benefit from a clearer explanation of what is the function of the supplemented bacterial RNA step 3.
ANSWER: We appreciate your comment. We clarified in the description of Figure 2d (step 3) that bacterial RNA was used as a heterologous source of non-specific RNA to assess whether the observed interaction between snR191 and PIP2 is specific. By supplementing with bacterial RNA, we sought to compete for potential non-specific binding sites on the membrane, reinforcing the specificity of the interaction of snR191 with PIP2. This information has also been incorporated into the legend of Figure 2d (lines 163-164) and the results (lines 144-145) for clarity.
- In Figure 2e, what does “Input" represents in the gel image? Is it the total RNA before the assay? The bar graph needs clear units on the y-axis (NU is defined, but it might be clearer with "Relative Quantity" or similar, along with NU). it states "p−value<0.05, ns= not significant" for the statistical analysis here, please ensure that the statistical test (t-test) and significance levels are clearly indicated for all comparisons shown in the graph. Please correct the inconsistent capitalization of "Pi(4,5)P2" in Figure.
ANSWER: Thank you for your comment. We have updated the legend of Figure 2e to clarify that “Input” refers to the total RNA used prior to the interaction assay. Additionally, the Y-axis of the bar graph has been modified to include the unit “Relative amount (NU)”, and “NU” has been defined in the figure legend. Information regarding the statistical analysis has also been added, indicating that a Student’s t-test was used with specific significance levels (p<0.05, ns= not significant). Finally, we have consistently corrected “Pi(4,5)P2) throughout the figure and its legend (Lines 168 – 173).
- Please clarify the selection process for snR191 among the four cerevisiae snRNAs shown in Figure 3d as being associated with PIP2, providing context for its specific inclusion.
ANSWER: We appreciate your comment. We have addressed and clarified this point in the Abstract.
- Please correct the inconsistent values in Figure 4e and the figure legends. The binding affinity value is very high. While docking scores can be high, it would be beneficial to provide context or a brief interpretation of this value in relation to known RNA-ligand interactions, or mention that these are theoretical predictions that require experimental validation of affinity.
ANSWER: We appreciate the comment. Therefore, we have added the phrase "Although the predicted theoretical affinity of snR191-PIP2 was very high (Figure 4e) due to the docking program used, it needs to be experimentally validated" (Lines 212- 213) to the results section.
- Please rephrase the frank statement in lines 287-290. While acknowledging the study's limitations, it would be more impactful to propose specific experimental approaches (e.g., mutational analysis combined with binding assays, isothermal titration calorimetry) that would validate the predicted interaction mode, demonstrating a comprehensive understanding of the necessary next steps.
ANSWER: We appreciate this suggestion. We have considered your observation and explored several experimental approaches that could validate this interaction. In lines 311 - 325, strategies such as mutation analysis, isothermal titration calorimetry, and surface plasmon resonance are proposed, which will allow us to verify the predictions made and experimentally validate the proposed model.
- Typographical errors and grammatical issues: description of Table S1 in line 174-175, title of Figure 3d, and hypos “Figure 64” in line 197, "Fig. 1d" in line 248 and "Fig. 1e" in line 249, “Figure 6c” in line 287.
ANSWER: We have corrected the noted typographical and grammatical errors. In particular:
- The description in Table S1 has been reworded for clarity.
- The title of Figure 3d has been corrected to read: "Multiple alignment of nucleotide sequences of four snoRNAs, with LINK-A as reference."
- The error in “Figure 64”, “Figure 1e”, “Figure 1d”, and Figure “6C” has been corrected.
Round 2
Reviewer 3 Report
Comments and Suggestions for Authors
Thank you for the authors' detailed reply and evident determination to improve the manuscript. I have no further comments.